# End-to-End Deep Learning Approach for Perfusion Data: A Proof-of-Concept Study to Classify Core Volume in Stroke CT

**DOI:** 10.3390/diagnostics12051142

**Published:** 2022-05-05

**Authors:** Andreas Mittermeier, Paul Reidler, Matthias P. Fabritius, Balthasar Schachtner, Philipp Wesp, Birgit Ertl-Wagner, Olaf Dietrich, Jens Ricke, Lars Kellert, Steffen Tiedt, Wolfgang G. Kunz, Michael Ingrisch

**Affiliations:** 1Department of Radiology, University Hospital, LMU Munich, 81377 Munich, Germany; paul.reidler@med.uni-muenchen.de (P.R.); matthias.fabritius@med.uni-muenchen.de (M.P.F.); balthasar.schachtner@med.uni-muenchen.de (B.S.); philipp.wesp@med.uni-muenchen.de (P.W.); olaf.dietrich@med.uni-muenchen.de (O.D.); jens.ricke@med.uni-muenchen.de (J.R.); wolfgang.kunz@med.uni-muenchen.de (W.G.K.); michael.ingrisch@med.uni-muenchen.de (M.I.); 2Comprehensive Pneumology Center (CPC-M), German Center for Lung Research (DZL), 81377 Munich, Germany; 3Department of Diagnostic Imaging, The Hospital for Sick Children, University of Toronto, Toronto, ON M5G 1X8, Canada; birgitbetina.ertl-wagner@sickkids.ca; 4Department of Neurology, University Hospital, LMU Munich, 81377 Munich, Germany; lars.kellert@med.uni-muenchen.de; 5Institute for Stroke and Dementia Research, University Hospital, LMU Munich, 81377 Munich, Germany; steffen.tiedt@med.uni-muenchen.de

**Keywords:** CT perfusion, stroke, deep learning, contrast-enhanced perfusion imaging, convolutional neural networks, end-to-end modeling

## Abstract

(1) Background: CT perfusion (CTP) is used to quantify cerebral hypoperfusion in acute ischemic stroke. Conventional attenuation curve analysis is not standardized and might require input from expert users, hampering clinical application. This study aims to bypass conventional tracer-kinetic analysis with an end-to-end deep learning model to directly categorize patients by stroke core volume from raw, slice-reduced CTP data. (2) Methods: In this retrospective analysis, we included patients with acute ischemic stroke due to proximal occlusion of the anterior circulation who underwent CTP imaging. A novel convolutional neural network was implemented to extract spatial and temporal features from time-resolved imaging data. In a classification task, the network categorized patients into small or large core. In ten-fold cross-validation, the network was repeatedly trained, evaluated, and tested, using the area under the receiver operating characteristic curve (ROC-AUC). A final model was created in an ensemble approach and independently validated on an external dataset. (3) Results: 217 patients were included in the training cohort and 23 patients in the independent test cohort. Median core volume was 32.4 mL and was used as threshold value for the binary classification task. Model performance yielded a mean (SD) ROC-AUC of 0.72 (0.10) for the test folds. External independent validation resulted in an ensembled mean ROC-AUC of 0.61. (4) Conclusions: In this proof-of-concept study, the proposed end-to-end deep learning approach bypasses conventional perfusion analysis and allows to predict dichotomized infarction core volume solely from slice-reduced CTP images without underlying tracer kinetic assumptions. Further studies can easily extend to additional clinically relevant endpoints.

## 1. Introduction

Acute ischemic stroke occurs when a blood clot interrupts the blood flow (perfusion) to the brain, most commonly in a supplying artery—this causes cell death in the hypoperfused areas [1]. Historically, cerebral perfusion imaging was performed using positron emission tomography using radioactive labeled oxygen to determine oxygen fraction and cerebral metabolic rate for oxygen or single photon emission computed tomography. However, logistics and application of radiotracers made both modalities unfeasible for the emergency setting. Today, computed tomography perfusion (CTP) is the most frequently used method to classify the salvageable brain tissue (penumbra) from the irreversibly damaged core in order to support clinical decision-making [2,3,4].

CTP is based on consecutive sampling of cerebral tissue attenuation after intravenous bolus injection of an iodinated contrast agent. A time-attenuation curve in every voxel represents the passage of the contrast agent through the brain in the reconstructed 4D image. After conversion to concentration, tracer-kinetic analysis aims to quantitatively evaluate the time-concentration curves by estimating perfusion parameters, e.g., cerebral blood volume, cerebral blood flow, time to peak, and mean transit time [5]. The most common approach uses deconvolution: An arterial input function (AIF) is determined in a large feeding artery and the time-concentration curves are deconvolved voxel-wise with the AIF to estimate perfusion parameters [6,7,8].

Radiologists as human experts then examine the generated perfusion parameter maps to detect hypoperfused areas, i.e., penumbra and core, and decide among treatment options. In the setting of acute ischemic stroke, CTP can help to identify patients who have a large penumbra and a small core, as they are likely to have a favorable response to reperfusion therapies [9,10]. Additionally, it was shown that CTP can help identify stroke mimics like epilepsy [11] and improves the detection performance for peripheral ischemia with often minor clinical symptoms, which is paramount for future therapy concepts on medium vessel occlusion [12]. However, availability and usage of advanced stroke imaging methods, including CTP, vary considerably among sites and geographical areas, with only around half of centers using those methods frequently [13].

The value of convolutional neural networks (CNN) has been demonstrated for a variety of medical imaging tasks, e.g., image reconstruction, object detection, segmentation, or classification [14,15,16,17,18,19]. The reason for the success of these systems is based on the capability of CNNs to learn data-driven features from pixels directly. Multiple nonlinear processing layers produce a high-level representation of features in images. Consequently, CNN-based approaches were proposed and applied to perfusion imaging analysis. For dynamic contrast-enhanced (DCE) MRI for example, CNN models were developed to estimate perfusion parameters maps directly from the data without the requirement for a standard deconvolution process [20,21]. A recent study [22] proposed a voxelwise prediction of infarct status from stroke CTP, but with the use of additional clinical and tracer-kinetic related data.

Commercial CTP analysis approaches often require domain expertise—e.g., by localizing, verifying, or correcting vessels for the measurement of arterial input functions. Uncertainties, e.g., induced by partial volume effects, propagate into the calculation of perfusion parameter maps via tracer kinetic modeling and ultimately in the process of clinical decision-making. Deep learning, on the other hand, may enable the direct prediction of clinical endpoints from complex imaging data with minimal user input. Starting from the baseline model developed in this study, more complex models can be adapted to relevant clinical endpoints in acute ischemic stroke, e.g., grade of disability or quality of life. For ischemic stroke, the most widely applied measure for neurological outcome uses the modified Rankin scale at day 90 after stroke. Clinical endpoints are likely associated with subtle patterns in the data and development of deep learning models generally requires large datasets [23]. In a proof-of-concept study, we, therefore, aimed to categorize patients into small or large core with an end-to-end deep learning approach for slice-reduced CT perfusion data.

## 2. Materials and Methods

### 2.1. Study Population, Image Acquisition and Core Volumetry

In this retrospective study, we included a training cohort (*n* = 217) (Figure 1) from among 234 consecutive acute ischemic stroke patients with available raw CTP data from a prospectively acquired cohort (German Stroke Registry, NCT03356392). All patients were treated with endovascular mechanical thrombectomy at our institution. We excluded patients with inconsistent CTP images that did not comply with a standardized time resolution of 1.5 s or a scan duration of 48 s. Patients underwent CTP on admission using a SOMATOM Definition Force, AS+, or Flash CT scanner (Siemens Healthineers, Forchheim, Germany). Automated calculation of ischemic core was performed using the CT vendor’s proprietary software (syngo Neuro Perfusion CT; Siemens Healthineers, Forchheim, Germany), which applies a threshold cerebral blood volume of <1.2 mL/100 mL. Median core volume was calculated and used as threshold value for binary classification: Small core < median core, large core > median core. As a proof-of-principle, we used median core volume to ensure a balanced split of the training data, in contrast to a fixed threshold value of 70 mL, which can sometimes be found in the literature [24].

For independent validation, we included a second external test cohort (*n* = 23) from among 63 patients of the external, publicly available ISLES 2018 challenge data set [25,26]. We excluded patients with inconsistent CTP images that could not be interpolated to the standardized time resolution of 1.5 s and a scan duration of 48 s. The challenge data include core segmentations, which were used to calculate the ischemic core volume. For this purpose, the number of voxels in the segmentation was multiplied by the voxel dimensions to get an estimated core volume in ml. The median core volume of the training dataset was used as threshold value for binary classification. Figure 1 shows a detailed flowchart of patient selection for both cohorts.

### 2.2. Preprocessing, Batch Generation, and Data Augmentation

Internal and external datasets were both preprocessed in two steps. First, two axial slices covering the middle cerebral artery territory (basal ganglia and supraganglionic level) based on the Alberta stroke program early CT score (ASPECTS) [27] regions were selected by radiologists. In a second fully automated step, the selected slices were resized and interpolated to 128 × 128 pixels in-plane resolution with 200 × 200 mm^2^ length. All slices along the time axis were co-registered to the first slice at t = 0 to reduce motion artifacts. All data were processed using custom Python (version 3.8.5) [28] scripts including the publicly available packages SimpleITK (version 2.0.2) [29] and Scikit-learn (version 0.23.2) [30].

During training, a custom batch generator returned a random subset of samples (batch size = 12) from the complete dataset and normalized each batch to zero mean and unit variance. Online data augmentation was applied to each batch in the form of random rotation in the range of (−15°, 15°), xy-shift (−10 pixel, 10 pixel), and vertical flip (True, False) before passing it on to the network.

### 2.3. Network Architecture

The proposed network architecture was implemented in Python and TensorFlow (version 2.3.0) [31] and is illustrated in Figure 2. It consists of two submodels with identical architecture for each of the selected axial CTP slices. The standardized and augmented 2D+t input images were fed into each submodel and processed through the pipeline to extract spatial and temporal features. The resulting features are concatenated, passed through a fully connected dense layer, and classified (Figure 2 top).

Figure 2 (bottom, zoomed-in) displays a detailed demonstration of the feature extraction part. For spatial feature extraction, each 2D image on the time axis is fed into a VGG19 model [32], pretrained on 2D image net data [33]. The weights are shared across all timepoints within a “TimeDistributed” framework. The resulting 32 × 512 feature matrix is passed on to the temporal feature extraction step. The temporal feature extraction consists of a 1D convolution with three filters followed by a max pooling layer and is divided into a global and a local pathway. In the global pathway, the 1D convolution is performed with kernel size 11, in the local pathway with kernel size 3. This ensures that the model can capture both smaller and larger changes along the time course. The resulting feature vectors are concatenated and passed on to a dense layer with 32 units. Classification is performed using a sigmoid layer. The source code is made publicly available on the development platform Github (https://github.com/AndreasMittermeier/stroke-perfusion-CNN (accessed on 7 March 2022)).

### 2.4. Training, Validation and Testing

The proposed network was trained, validated, and tested using all included patients from the training cohort within a 10-fold cross-validation (CV). The dataset was randomly split into ten folds according to an 8:1:1 ratio of training, validation, and test. Eight folds were used for training the network. The number of training epochs was set to 500. The validation fold was used to evaluate the model after each epoch and stop training once the validation loss stopped decreasing for 200 epochs (patience = 200). After the last epoch, the test fold was evaluated by the model with the best weights, i.e., the weights which yielded the lowest validation loss. After ten CV iterations, each fold was used for unbiased testing once. The area under the receiver operating characteristic curve (ROC-AUC) was used as evaluation metric. Mean and standard deviation (SD) of ROC-AUC values were reported for the 10-fold CV.

In addition, we performed an ablation study on the effect of the local and global temporal feature extraction. To this end, we trained two reduced models using the (i) local feature extractor alone and using the (ii) global feature extractor alone. Training and evaluation on the test folds was performed in the same CV approach as described above. Mean and SD of the ROC-AUC values were reported and compared with those of the full model.

To evaluate the independent test cohort, an ensemble method was used. The final model was constructed by averaging the predictions from the ten models trained in the CV. The final model was applied to the independent test cohort and the ROC-AUC was reported.

## 3. Results

Two hundred seventeen patients were included in the training cohort and 23 patients in the independent test cohort. Median core volume for the training cohort was 32.4 mL which yields, per definition, a balanced class split. Applying this threshold to the independent test data resulted in 12 patients with large core volume and 11 patients with small core volume. Training duration for the 10-fold CV was in the range of 24 h on a local workstation (NVIDIA GeForce RTX 2070 Super) with online data augmentation and batch-wise data standardization. Evaluation and prediction of unseen data were of the order of a few seconds.

Figure 3 shows the ROC curves for the test folds within the 10-fold CV. The mean ROC curve is overlaid in blue, an interval of ±1 SD is shaded in grey, and the dashed line represents random guessing. The mean (SD) ROC-AUC over 10 folds was 0.72 (0.10) for the test folds. In comparison, mean (SD) ROC-AUC for the validation folds, which were used to early stop training, was 0.75 (0.11). The averaged ensemble ROC-AUC for the independent test cohort selected from the ISLES 2018 challenge was 0.61, which is close to the ±1 SD interval of the test folds from the training data. All results are summarized in Table 1.

The ablation study showed a decrease in mean (SD) ROC-AUC values for the reduced models, summarized in Table 2. Using the global feature extractor alone resulted in a ROC-AUC of 0.63 (0.14) and using the local feature extractor alone resulted in a ROC-AUC of 0.65 (0.13), compared to the full model with ROC-AUC 0.72 (0.10).

## 4. Discussion

In this proof-of-concept study, we developed a novel end-to-end deep learning approach to bypass conventional perfusion analysis, which allows to directly categorize patients into small or large core from raw CTP data without tracer-kinetic assumptions. We demonstrated this approach on 217 patients with acute ischemic stroke by directly predicting dichotomized infarct core volume and showed that the model learned relevant spatial and temporal features purely from the data. The results of the ablation study demonstrate the advantage of combined local and global temporal feature extraction, as only the full model yields the best performance. This corroborates our understanding that both short-term effects (e.g., sharp peaks in concentration) and long-term effects (e.g., wash-out) add relevant information and must therefore both be considered in the model architecture. In this proof-of-concept approach, our model cannot be translated to clinical practice immediately, however, achieved good predictive performance on an inhouse dataset in a 10-fold CV approach and generalized to independent test data, showing the potential of end-to-end CT perfusion analysis.

The proposed deep learning model is based on a 2D approach that covers a reduced portion of the middle cerebral artery territory, represented by the ASPECTS regions. Arguably, a 3D approach would contain more relevant information, but whole-brain CT perfusion is not available in all primary stroke centers [34]. Using the 2D approach, we were able to include all possible data, especially the external test data, which consisted of two separate stacks of slices instead of whole-brain perfusion. We believe that this 2D approach is sufficient to prove the concept that spatial and temporal information can be extracted from CTP data to predict dichotomized core volume using deep learning.

In comparison to existing studies using deep learning for perfusion analysis in stroke CT [8], we focused on using the raw perfusion data solely. In contrast to voxelwise prediction of infarct status, the proposed model learned to predict dichotomized infarction core volume without taking additional parameters into account. While additional parameters like treatment information may be beneficial, additional user-provided information, such as a manually selected arterial input function, requires input from expert users and introduces user dependency. Our proposed model learned the link between perfusion input and tissue response purely based on the data and is free from tracer-kinetic assumptions.

Imaging-derived parameters play a crucial role in clinical decision-making in the setting of acute ischemic stroke. Foremost, CTP-derived ischemic core volume has become one of the key parameters in the decision for mechanical thrombectomy in the extended time window [35,36]. Currently, there is no consensus on the use of CTP-parameters for core/penumbra estimation. While software in large clinical trials used relative CBF thresholds (RAPID), other software relies on MTT (Philips Brain CT perfusion) or, in our case, on CBV. As relevant differences have been shown between vendors, our approach needs further validation for other CTP analysis thresholds [37]. In the present proof-of-concept study, we predicted dichotomized ischemic core volume as a simplified endpoint. Given sufficient training data, this approach can easily be generalized to more complex labels in future studies, such as impairment after discharge. The underlying relationship is harder to learn and may require incorporating additional clinical parameters into the model. Such deep-learning-based approaches may, therefore, be used to predict complications and even chronic functional outcomes in order to guide clinical management in and beyond the acute stroke phase.

The present study is not without limitations. First, the sample size of 217 training datasets is small for the complex problem of directly predicting an imaging-derived parameter from raw data and validation with a larger dataset is needed. To this end, dichotomized median core volume was chosen (i) to cast the problem as classification approach and (ii) to provide a balanced group distribution for the training dataset. This preliminary work may lay the groundwork for future studies to examine more clinically relevant endpoints, such as grade of disability or quality of life. Nevertheless, as proof of concept, our model achieved good results, which were validated on external test data. The performance gap between model predictions for in-house and external test data is likely due to differences in data quality and core labeling. Second, the number of perfusion timepoints of the CTP images was fixed for the model input, but appropriate interpolation could solve varying temporal resolutions.

## 5. Conclusions

In this proof-of-concept study, the proposed end-to-end deep learning approach bypasses conventional perfusion analysis and allows training a model that predicts dichotomized infarction core volume solely from slice-reduced CTP images without underlying tracer kinetic assumptions. Further studies can easily extend to additional clinically relevant endpoints.

## Figures and Tables

**Figure 1 diagnostics-12-01142-f001:**
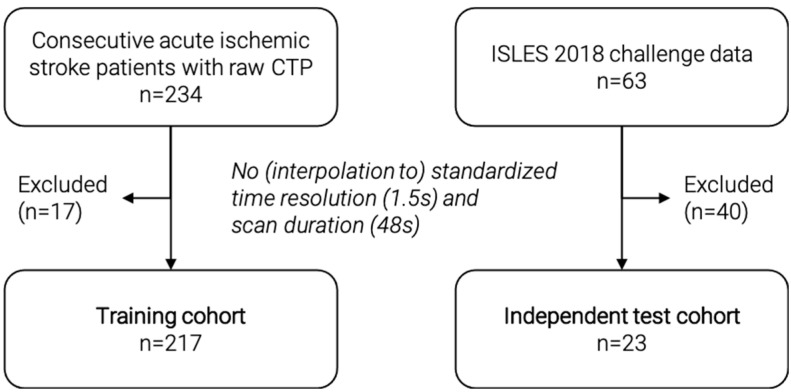
Flowchart of patient selection for the training and independent test cohort. CTP = CT perfusion.

**Figure 2 diagnostics-12-01142-f002:**
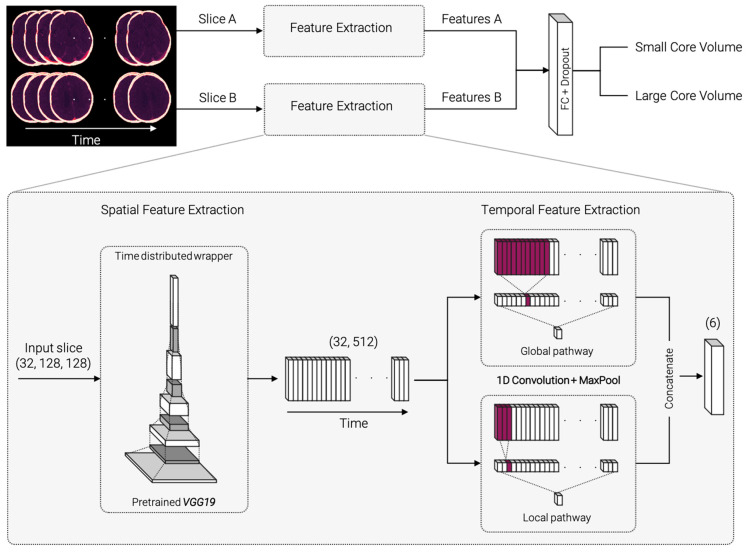
Model architecture overview and detailed, zoomed-in view of the spatial and temporal feature extraction process. The selected slices A and B are fed into identical submodels for spatial and temporal feature extraction. Spatial feature extraction consists of identical, pretrained VGG19 networks for each timepoint of the input images. The resulting feature vector is passed on to the temporal feature extraction. 1D convolutions with two different kernel sizes are carried out in a global and local pathway. The extracted features A and B for both submodels are concatenated, fully connected (FC), and classified.

**Figure 3 diagnostics-12-01142-f003:**
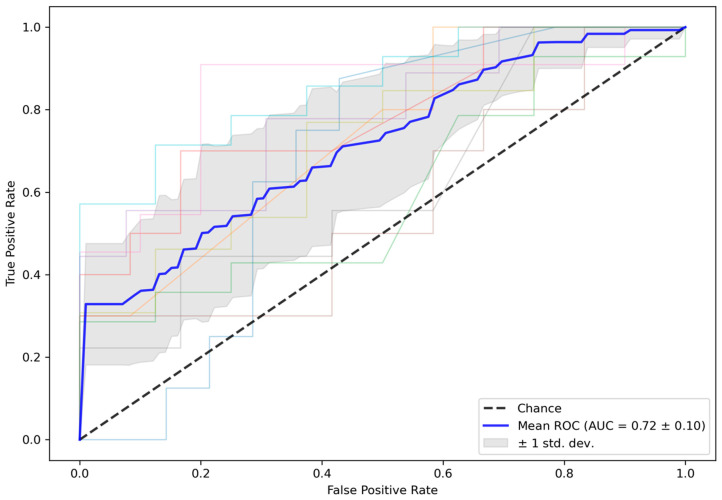
ROC curves for test data in the 10-fold CV. Mean (SD) ROC-AUC for the CV test folds was 0.72 (0.10). CV = cross-validation, ROC-AUC = area under the receiver operator characteristics curve.

**Table 1 diagnostics-12-01142-t001:** Mean (SD) ROC-AUC of the final model for validation and test folds during CV and for the external test cohort. SD = standard deviation, CV = cross-validation, ROC-AUC = area under the receiver operator characteristics curve.

Validation Folds	Test Folds	Independent Test Cohort
0.75 (0.11)	0.72 (0.10)	0.61

**Table 2 diagnostics-12-01142-t002:** Mean (SD) ROC-AUC for the test folds during CV of the full model and the reduced models within the ablation study setting. SD = standard deviation, CV = cross-validation, ROC-AUC = area under the receiver operator characteristics curve.

Full Model	Global Feature Extractor Alone	Local Feature Extractor Alone
0.72 (0.10)	0.63 (0.14)	0.65 (0.13)

## Data Availability

Validation data available in a publicly accessible repository. The data of the ISLES 2018 challenge used as validation data in this study are openly available at https://www.smir.ch/ISLES/Start2018 (accessed on 2 May 2021).

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
