# Peer review of "End-to-End Deep Learning Approach for Perfusion Data: A Proof-of-Concept Study to Classify Core Volume in Stroke CT"

_diagnostics, 2022, doi:10.3390/diagnostics12051142_

Round 1

Reviewer 1 Report

In this work by Mittermeier et al authors evaluate the issue concerning the perfusion analysis in stroke. Authors refer to Computed Tomography perfusion and base their observations on 217 patients and 23 in the independent study cohort. Several points should be addressed before further processing:

  1. In the introduction authors should refer to other methods quantifying cerebral hypoperfusion e.g. SPECT/PET
  2. As ischaemic stroke is a major challenge in the treatment of general neurology, authors could elaborate on the accessibility and perspectives of CTP
  3. In the context of the limitations authors should stress in the title and throughout the more preliminary character of the study. Perhaps a short report would be a more fitting article type.

Author Response

Comments and Suggestions for Authors

In this work by Mittermeier et al authors evaluate the issue concerning the perfusion analysis in stroke. Authors refer to Computed Tomography perfusion and base their observations on 217 patients and 23 in the independent study cohort. Several points should be addressed before further processing:

Point 1: In the introduction authors should refer to other methods quantifying cerebral hypoperfusion e.g. SPECT/PET

Response 1: We agree with the reviewer and reworked the respective parts of the first paragraph of the introduction which now reads:

Historically, cerebral perfusion imaging was performed using positron emission tomography using radioactive labeled oxygen to determine oxygen extraction fraction and cerebral metabolic rate for oxygen or single photon emission computed tomography. However logistics and application of radiotracers made both modalities unfeasible for the emergency setting. Today, computed tomography perfusion (CTP) is the most frequently used method to classify the salvageable brain tissue (penumbra) from the irreversibly damaged core in order to support clinical decision making [2–4].

Point 2: As ischaemic stroke is a major challenge in the treatment of general neurology, authors could elaborate on the accessibility and perspectives of CTP

Response 2: We thank the reviewer for this considerate comment which is also touched by reviewer 2 point 2. We expanded the respective parts of the introduction which now discusses availability and usage of CTP for therapy decisions:

In the setting of acute ischemic stroke, CTP can help to identify patients who have a large penumbra and a small core, as they are likely to have a favorable response to reperfusion therapies [9,10]. Additionally, it was shown that CTP can help identify stroke mimics like epilepsy [11] and improves the detection performance for peripheral ischemia with often minor clinical symptoms, which is paramount for future therapy concepts on medium vessel occlusion [12]. However, availability and usage of advanced stroke imaging methods, including CTP, vary considerably among sites and geographical areas, with only around half of centers using those methods frequently [13].

Point 3: In the context of the limitations authors should stress in the title and throughout the more preliminary character of the study. Perhaps a short report would be a more fitting article type.

Response 3: We agree with the reviewer that it is important to emphasize the preliminary nature of this study. Therefore, we have chosen to include “proof-of-concept study” in our title, so that readers are aware of the qualities of the study. We think that further use of “preliminary” in the title would raise the reader´s concern about the presented data, which, in actuality, was extensively analyzed using multiple methods and cohorts. Throughout the text, we clarified the respective passages and adapted the wording to reflect the proof-of-concept character. To further emphasize these aspects, the first paragraph of the discussion now reads:

In this proof-of-concept study, we developed a novel end-to-end deep learning approach to bypass conventional perfusion analysis which allows to directly categorize patients into small or large core from raw CTP data without tracer-kinetic assumptions.

In the present study, we developed and validated a novel concept for CTP analysis using CNNs to extract spatial and temporal information from time-resolved stroke imaging data. This includes development and implementation of new analysis strategies in a deep learning framework, as well as the application and validation on in-house training and external test data. Due to the extensive data and analysis in the manuscript, we consider an original article format justified, however we would accept transition to a shorter format at the editor's discretion.

Reviewer 2 Report

I read with real pleasure your work (End-to-end deep learning approach for perfusion data: a proof-of-concept study to classify core volume in stroke CT), written in the wiev of a simplification of the usual process of patient’s categorization about CTP.

The possibility of apply CNNs to stroke’s imaging- in this particular case, to CTP- is a very good idea.

Discussion is very focused on the problem, concise and interesting.

There’s a big limit of a simple 2D model of the territory of the middle cerebral artery, and I agree with you that we can push our boundaries including the whole brain- well, this could be the starting point of your new work!!!

I have some clinical disagreement:

1) you rightly wrote that CTP-derived ischemic core volume has become one of the key parameters in the decision for mechanical thrombectomy in the extended window; but we have to underline that in many other situations CTP is useful, as the discrepancy between radiolocal and clinical signs. 2) Not so many stroke’s centers have the possibility to obtain CTP studies 24h, 7days a week, and a goog knowledge of other factors involved in this process is something that you have to write ( I suggest a reference: Alexandre AM, Pedicelli A, Valente I, Scarcia L, Giubbolini F, D'Argento F, Lozupone E, Distefano M, Pilato F, Colosimo C. May endovascular thrombectomy without CT perfusion improve clinical outcome? Clin Neurol Neurosurg. 2020 Nov;198:106207. doi: 10.1016/j.clineuro.2020.106207. Epub 2020 Sep 7. PMID: 32950754.) 3) Your sample is too small for a retrospective stroke’s paper 4) CTP and all radiological findings must have a meaning thus in relationship with clinical correlates, such as modified Rankin scale; 5) Quality of images should be improve. 6) Finally I would like you to discuss CTP parameters, since there is no consensus on this, and different parameters are used.

Author Response

Comments and Suggestions for Authors

I read with real pleasure your work (End-to-end deep learning approach for perfusion data: a proof-of-concept study to classify core volume in stroke CT), written in the wiev of a simplification of the usual process of patient’s categorization about CTP.

The possibility of apply CNNs to stroke’s imaging- in this particular case, to CTP- is a very good idea.

Discussion is very focused on the problem, concise and interesting.

There’s a big limit of a simple 2D model of the territory of the middle cerebral artery, and I agree with you that we can push our boundaries including the whole brain- well, this could be the starting point of your new work!!!

We thank the reviewer very much for this encouraging and inspiring comment. Indeed, we want to lay the groundwork for the more complex scenario of end-to-end 3D whole brain CTP analysis with this proof-of-concept study.

I have some clinical disagreement:

Point 1: you rightly wrote that CTP-derived ischemic core volume has become one of the key parameters in the decision for mechanical thrombectomy in the extended window; but we have to underline that in many other situations CTP is useful, as the discrepancy between radiolocal and clinical signs. 

Response 1: We completely agree with the reviewer that CTP is useful in many clinical scenarios. We expanded the introduction and included a statement to clarify the additional use of CTP (please see also Response 2 for additional changes to the manuscript):

In the setting of acute ischemic stroke, CTP can help to identify patients who have a large penumbra and a small core, as they are likely to have a favorable response to reperfusion therapies [9,10]. Additionally, it was shown that CTP can help identify stroke mimics like epilepsy [11] and improves the detection performance for peripheral ischemia with often minor clinical symptoms, which is paramount for future therapy concepts on medium vessel occlusion [12]. However, availability and usage of advanced stroke imaging methods, including CTP, vary considerably among sites and geographical areas, with only around half of centers using those methods frequently [13].

Point 2: Not so many stroke’s centers have the possibility to obtain CTP studies 24h, 7days a week, and a goog knowledge of other factors involved in this process is something that you have to write ( I suggest a reference: Alexandre AM, Pedicelli A, Valente I, Scarcia L, Giubbolini F, D'Argento F, Lozupone E, Distefano M, Pilato F, Colosimo C. May endovascular thrombectomy without CT perfusion improve clinical outcome? Clin Neurol Neurosurg. 2020 Nov;198:106207. doi: 10.1016/j.clineuro.2020.106207. Epub 2020 Sep 7. PMID: 32950754.) 

Response 2: Here, the reviewer touches a critical point which was also addressed by reviewer 1 point 2. We elaborated further on the availability and usage of CTP in the introduction and included the suggested reference [10]:

In the setting of acute ischemic stroke, CTP can help to identify patients who have a large penumbra and a small core, as they are likely to have a favorable response to reperfusion therapies [9,10]. Additionally, it was shown that CTP can help identify stroke mimics like epilepsy [11] and improves the detection performance for peripheral ischemia with often minor clinical symptoms, which is paramount for future therapy concepts on medium vessel occlusion [12]. However, availability and usage of advanced stroke imaging methods, including CTP, vary considerably among sites and geographical areas, with only around half of centers using those methods frequently [13].

Point 3: Your sample is too small for a retrospective stroke’s paper 

Response 3: From a clinical perspective, we completely agree with the reviewer. However, our results indicate that our sample size was sufficient to learn the classification problem at hand in our proof-of-concept study. We reworked the discussion to that regard which now reads:

The present study is not without limitations. First, the sample size of 217 training datasets is small for the complex problem of directly predicting an imaging derived parameter from raw data and validation with a larger dataset is needed.

Also, we underlined the preliminary nature of this work now in the first sentence of the discussion: 

In this proof-of-concept study, we developed a novel end-to-end deep learning approach to bypass conventional perfusion analysis which allows to directly categorize patients into small or large core from raw CTP data without tracer-kinetic assumptions.

Point 4: CTP and all radiological findings must have a meaning thus in relationship with clinical correlates, such as modified Rankin scale; 

Response 4: We thank the reviewer for pointing this out and want to emphasize that we are well aware of the meaningfulness of clinical endpoints. However, in this proof-of-concept study we tried to build a baseline model which can be adapted to that regard. To clarify this, we reworked the last paragraph of the introduction accordingly:

Starting from the baseline model developed in this study, more complex models can be adapted to relevant clinical endpoints in acute ischemic stroke, e.g., grade of disability or quality of life. For ischemic stroke, the most widely applied measure for neurological outcome uses the modified Rankin scale at day 90 after stroke.

Also, we present this aspect in the limitation section: 

This preliminary work may lay the groundwork for future studies to examine more clinically relevant endpoints, such as grade of disability or quality of life. 

Point 5: Quality of images should be improve. 

Response 5: We thank the reviewer for pointing this out. We assume that the lower resolution of the figures is only a problem with the preliminary manuscript version as we uploaded separate, high-resolution images during the submission process.

Point 6: Finally I would like you to discuss CTP parameters, since there is no consensus on this, and different parameters are used.

Response 6: We are aware of the discussion and reworked the text concerning the use of CTP parameters and elaborated on differences between vendors; the third paragraph of the discussion now reads:

Foremost, CTP-derived ischemic core volume has become one of the key parameters in the decision for mechanical thrombectomy in the extended time window [35,36]. Currently, there is no consensus on the use of CTP-parameters for core/penumbra estimation. While software in large clinical trials used relative CBF thresholds (RAPID), other software relies on MTT (Philips Brain CT perfusion) or, in our case, on CBV. As relevant differences have been shown between vendors, our approach needs further validation for other CTP analysis thresholds [37].

Round 2

Reviewer 1 Report

I do not have further comments.